# CONTROLLABLE PARETO TRADE-OFF BETWEEN FAIRNESS AND ACCURACY

## ABSTRACT

The fairness-accuracy trade-off is a fundamental challenge in machine learning. While simply combining the two objectives can result in mediocre or extreme solutions, multi-objective optimization (MOO) could potentially provide diverse trade-offs by visiting different regions of the Pareto front. However, MOO methods usually lack precise control of the trade-offs. They rely on the full gradient per objective and inner products between these gradients to determine the update direction, which can be prone to large data sizes and the curse of dimensionality when training millions of parameters for neural networks. Moreover, the trade-off is usually sensitive to naïve stochastic gradients due to the imbalance of groups in each batch and the existence of various trivial directions to improve fairness. To address these challenges, we propose "**C**ontrollable **P**areto **T**rade-off (CPT)" that can effectively train models performing different trade-offs defined by reference vectors. CPT begins with a correction stage that solely approaches the reference vector and then includes the discrepancy between the reference and the two objectives as the third objective in the rest training. To overcome the issues caused by high-dimensional stochastic gradients, CPT (1) uses a moving average of stochastic gradients to determine the update direction; and (2) prunes the gradients by only comparing different objectives' gradients on the critical parameters. Experiments show that CPT can achieve a higher-quality set of diverse models on the Pareto front performing different yet better trade-offs between fairness and accuracy than existing MOO approaches. It also exhibits better controllability and can precisely follow the human-defined reference vectors.

## 1 INTRODUCTION

As machine learning (ML) plays a significant role in automated decision-making systems, the fairness of ML models over different groups becomes a critical concern in practical applications. Unfairness or bias in machine learning systems can manifest in various ways and across different domains, e.g., gender bias in hiring algorithms, racial bias in criminal justice, and biased recommendations in online platforms (Zhao et al., 2017). There are two main sources of unfairness in supervised machine learning (Liu & Vicente, 2022). First, the data used to train the model are collected from humans (or automated agents developed by humans), which may contain inherent biases, making it challenging to obtain unbiased predictions through standard learning processes. Second, since ML aims to make predictions as accurate as possible, the training process may rely on biased features and sacrifice fairness to achieve better accuracy. Hence, investigating the fairness-accuracy trade-off in machine learning is highly worthwhile.

Current bias mitigation methods can be divided into three categories (Hort et al., 2022; Friedler et al., 2019): (1) Pre-processing, which aims to remove bias from the training data and prevent it from affecting ML models (Calmon et al., 2017); (2) In-processing, which mitigates biases during the training process (Zafar et al., 2017). The most intuitive strategy might be minimizing a linear combination of fairness and task loss (Roy & Ntoutsi, 2022), i.e., linearization. Another strategy is constrained optimization, which minimizes the task loss Cheng et al. (2022) under a fairness constraint; (3) Post-processing, which aims at reducing bias of trained ML models (Pleiss et al., 2017). Although the above methods are developed to balance fairness and accuracy, it is still an open challenge for them to precisely control and customize the trade-off. Specifically, the "optimal"

models with different trade-offs can be defined by the Pareto frontier, which is a set of equilibrium on which one cannot improve an objective without degrading another.

While the Pareto frontier of fairness and accuracy can be highly complicated and contains rich solutions performing different trade-offs, linearization with different weights cannot guarantee to visit all of them and in the worst case, it may only end up with models optimized for single objectives (§4.7.4 of Boyd & Vandenberghe (2004)). Moreover, the conflicts between objectives or constraints can limit the exploration of diverse solutions on the Pareto frontier. These are practical challenges for fairness-accuracy trade-off because there usually exist many directions to trivially improve the fairness with accuracy degradation (e.g., random predictions independent of the input can achieve the best fairness and there are many directions leading to random predictions). Multi-objective optimization (MOO) methods such as Multi-Gradient Descent Algorithm (MGDA) are able to converge to a Pareto equilibrium (Désidéri, 2012) by finding a common optimization direction in each step on which all objectives are improving or at least staying the same. In MGDA, the common direction is represented by a convex combination of gradients for all objectives (Sener & Koltun, 2018; Milojkovic et al., 2019). Moreover, MGDA has the potential to visit different regions of the Pareto frontier with the guidance of a predefined reference vector in the objective space, e.g., by introducing a new objective as the distance between the reference vector and the vector of objective values (Mahapatra & Rajan, 2020; Lin et al., 2019).

However, it is still challenging to directly apply MGDA to fairness-accuracy trade-off when training neural network models because: (1) MGDA relies on the full gradients of objectives to determine the common optimization direction, while stochastic gradient is more commonly used in training neural networks. Although stochasticity is important to model generalization and achieve high test accuracy, it may lead to a drift of the fairness loss since samples in a mini-batch might not cover all subgroups. (2) The inner products between gradients play an important role in determining the common optimization direction. However, when applied to train modern neural networks with millions of parameters, the curse of dimensionality might lead to less informative inner products reflecting the objective correlation. Moreover, many neural network parameters can be pruned without affecting the model performance but they together may contaminate the inner product and thus are detrimental to the search for the common descent direction. (3) It is challenging to control MGDA's optimization trajectory precisely following a pre-defined reference vector.

To overcome these challenges, we propose **C**ontrollable **P**areto Fairness-Accuracy **T**rade-off method (CPT). Our contribution can be summarized as follows:

- We utilize the moving average of stochastic gradients for each objective to approximate the full gradients used in MGDA for finding the common descent direction without missing subgroups.

- We prune the gradient per objective and use a joint mask to reduce all gradients' dimensionality so MGDA can estimate a more precise common descent direction out of the pruned gradients.

- Our experiments on Jigsaw dataset show that CPT, compared to a rich class of baselines, can better follow the reference vectors and find diverse Pareto solutions with different trade-offs, resulting in a better hypervolume on the test set.

The rest of the paper is organized as follows. Related works are presented in § 2. We describe our fairness-accuracy trade-off method in § 3. The experiment setting and results are in § 4, followed by the conclusion in § 5.

## 2 RELATED WORK

### 2.1 FAIRNESS-AWARE TRAINING

Recently, more and more attention has been paid to fairness-aware training in various research fields, such as natural language generation (Xu et al., 2022; Gupta et al., 2022), natural language processing (Zhao et al., 2017; Sheng et al., 2021), and multi-task learning (Roy & Ntoutsi, 2022; Oneto et al., 2019). Different approaches have been proposed for fairness-aware training. Commonly used methods include regularization, which adds a term to the loss function to penalize discrimination (Kamiran et al., 2010), and constraint optimization, which sets an upper bound for unfairness that cannot be breached during training (Kim et al., 2018; Cheng et al., 2022; Celis et al., 2019). A

more advanced approach is adversarial training, which simultaneously trains classification models and their discriminators (another classifier to predict sensitive attributes) (Lahoti et al., 2020; Beutel et al., 2017).

A significant fact for fairness-aware training is the trade-off between fairness and model performance. Huang & Vishnoi (2019) studies demographic parity and algorithmic stability from a theoretical perspective. Dutta et al. (2020) investigates the essential trade-off between fairness and accuracy metrics. Liu & Vicente (2022) treats fairness as another objective, which is defined by the correlation between sensitive attributes and prediction results, and optimizes fairness and accuracy simultaneously.

## 2.2 MULTI-OBJECTIVE OPTIMIZATION

Multi-objective optimization aims to find a set of solutions with different trade-offs instead of a single solution. Evolutionary search is a gradient-free method to solve MOO problem (Deb, 2011; Coello, 2006). However, the search could be time-consuming and an ideal trade-off can not be guaranteed. Another intuitive and seemingly easy solution is to transform the multi-objective problem into a single objective problem which is the weighted sum of all objectives (Ribeiro et al., 2014; Lin et al., 2019). But linearization could fail when objectives conflict with each other and finding the optimal weight can be laborious (Milojkovic et al., 2019). As for the gradient-based method, multi-gradient descent algorithm (MGDA) (Désidéri, 2012) applies gradient descent for solving MOO problems and is proved to converge to the Pareto Stationary solution. However, MGDA suffers from expensive computation and limited sparse solution issues. To address the first limitation, Stochastic MultiSubgradient Descent Algorithm (SMSGDA) (Poirion et al., 2017) is purposed and applied in fairness-accuracy trade-off problem (Liu & Vicente, 2022). As for the second limitation, a branch of work utilizes the reference vector to guide the optimization and generate diverse solutions. Pareto multi-task learning (PMTL) (Lin et al., 2019) divides the MOO problem into multiple subproblems according to the reference vectors and finds solutions in the regions that are close to reference vectors. Exact Pareto Optimal Search (EPO) (Mahapatra & Rajan, 2020) is able to follow the reference vector more precisely by optimizing uniformity, which is defined as the KL divergence between the weighted loss function and unity. However, how to address these issues in fairness-accuracy trade-off is still an open problem. Recently, Zhao et al. (2021) applies Monte Carlo tree search to divide the search space for efficient modeling and searching for the optimal solution. Navon et al. (2020) builds a hypernetwork to learn the Pareto front. Xiao et al. (2023) reduces the complexity and enhances the effectiveness of direction-oriented multi-objective optimization. In this paper, we present a novel method CPT, which applies MOO to achieve controllable trade-off between fairness and accuracy.

## 3 METHOD

### 3.1 PROBLEM DEFINITION

Multi-objective optimization optimizes different objectives simultaneously. It can be defined as:

$$\min_{\theta} \mathcal{L}(\theta) \triangleq (\mathcal{L}_1(\theta), \mathcal{L}_2(\theta), \cdots, \mathcal{L}_m(\theta))^\mathsf{T} \tag{1}$$

where $m$ is the number of objectives to optimize, $\theta$ indicates the parameters, $\mathcal{L}_i$ is the loss function of the $i$-th objective, and $\mathcal{L}(\theta)$ is the multi-objective loss function. The goal of multi-objective optimization is achieving Pareto equilibrium. In the following, we will introduce the definition of Pareto equilibrium, the general fairness-accuracy trade-off problem as well as the concept of common descent vector which lies a solid foundation for our method.

**Definition 1** Pareto equilibrium (stationary) for MOO

(1) Solution $\theta$ dominates solution $\hat{\theta}$ if $\forall i, \mathcal{L}_i(\theta) \leq \mathcal{L}_i(\hat{\theta})$ and $\mathcal{L}(\theta) \neq \mathcal{L}(\hat{\theta})$.

(2) A solution $\theta^*$ is called Pareto stationary if there is no solution $\theta$ that dominates $\theta^*$.

The Pareto set ($\mathcal{P}_\theta$) represents a set of solutions that are not dominated by others, where each solution achieves a certain trade-off between the objectives. $\mathcal{P}_\theta$ forms a boundary in the objective space, and any point inside this boundary represents a suboptimal solution because it can be improved in at

least one objective without degrading others. In this paper, we aim to find diverse trade-offs between fairness and accuracy near or on the Pareto frontier.

**Definition 2**  Fairness-Accuracy Trade-off

Given a dataset $D$, consisting of input features $X$, labels $Y$, and sensitive attributes $A$ (such as the demographic group information), we utilize CrossEntropy for accuracy loss and DiffEodd (Barocas et al., 2019) for fairness loss respectively.

$$\mathcal{L}_{acc} = -\frac{1}{N} \sum_{i=1}^{N} (y_i \log(F(x_i; \theta))) \tag{2}$$

$$\mathcal{L}_{fair} = \sum_{a \in \mathbf{A}} (|\text{TPR}_a - \text{TPR}_{\text{overall}}| + |\text{FPR}_a - \text{FPR}_{\text{overall}}|) \tag{3}$$

where $F(\theta)$ is the classifier with parameter $\theta$. Our goal is to train $F(\theta)$, so that it can perform well on classification task and make fair prediction for each subgroup. Then, the fairness-accuracy trade-off problem could be defined as:

$$\min_{\theta} \mathcal{L}(\theta) \triangleq (\mathcal{L}_{fair}(\theta), \mathcal{L}_{acc}(\theta))^{\mathsf{T}} \tag{4}$$

**Definition 3**  Common Descent Vector

When using the gradient-based optimization algorithm to solve the MOO problem, the common decent vector $\nabla_{\theta}\mathcal{L}(\theta)$ provides the direction for optimization and the distance to update along the direction. MGDA defines the common descent vector as the vector with minimum L2 norm in the convex hull of the gradient of each objective. When searching the direction for common descent vector, MGDA uses inner product of gradient vectors (more details in Appendix A.1). However, high dimension gradients could be dominated by noise, making the common descent vector calculated by MGDA imprecise. In this paper, we generalize MGDA to fairness-accuracy trade-off and purpose a novel method named CPT to fix the aforementioned issues.

### 3.2 Reference vector following

To better control the optimization, CPT utilizes reference vectors $\mathbb{V} = \{\boldsymbol{v}_1, \boldsymbol{v}_2, \cdots, \boldsymbol{v}_n\}$ to guide the optimization process. Different reference vectors set different constraints for the optimization process and lead to diverse trade-offs on the Pareto set. Then, the fairness-accuracy trade-off is converted into a constraint bi-objective optimization, which is defined in Eq.( 5). For each constraint bi-objective optimization problem, CPT intends to solve it in two stages: correction stage and MOO stage (Yang et al., 2021). In the correction stage, CPT applies single objective optimization to satisfy the constraint: $\Psi(\boldsymbol{l}, \boldsymbol{v}) < \psi$, where $\Psi$ is the Kullback–Leibler divergence between een reference vector and loss value vector, and $\psi$ is the predefined threshold$\in$.

$$\min_{\theta} \mathcal{L}(\theta), \text{s.t.} \Psi(\boldsymbol{l}, \boldsymbol{v}) \triangleq D_{KL} \left( \frac{\boldsymbol{l}}{\|\boldsymbol{l}\|_1} || \frac{\boldsymbol{v}}{\|\boldsymbol{v}\|_1} \right) \leq \psi, \tag{5}$$

where $\boldsymbol{v} = (v_{fair}, v_{acc})$ is the reference vector and $\boldsymbol{l} = (l_{fair}, l_{acc})$ is the vector for two objective loss values. When $\frac{v_{fair}}{v_{acc}} > 1$, we except $l_{acc}$ to be lower than $l_{fair}$, which means a preference for accuracy.

The correction stage provides a suitable starting point for the MOO stage that follows the reference vector $\boldsymbol{v}$. In the MOO stage, CPT solves the constraint bi-objective optimization by simultaneously optimizing three objectives including fairness loss $\mathcal{L}_{fair}$, accuracy loss $\mathcal{L}_{acc}$, and the KL divergence between reference vector and loss vector $\Psi(\boldsymbol{l}, \boldsymbol{v})$. Thus, the objective function for MOO stage can be written as:

$$\min_{\theta} \mathcal{L}(\theta) \triangleq \min_{\theta} (\mathcal{L}_{fair}(\theta), \mathcal{L}_{acc}(\theta), \Psi(\boldsymbol{l}, \boldsymbol{v}))^{\mathsf{T}} \tag{6}$$

### 3.3 Moving Average of Stochastic Gradient addresses Fairness loss drift

When optimizing a single objective, we usually employ a stochastic approach, where a subset of data is used to compute the mini-batch stochastic gradient. However, directly using stochastic gradient for MOO may not be a wise choice. First, the stochastic nature of the optimization process

introduces noise into the gradient, which could be misleading for calculating the common descent vector. Second, as one single mini-batch may not cover all the subgroups, the mini-batch fairness loss as well as its gradient could be inaccurate.

Inspired by SGD with momentum, which intends to stabilize the gradient during optimization, CPT keeps moving average gradients to approximate the whole gradients of objective functions. This method smooths the gradient of each objective before calculating the common descent vector, which leads to a more precise weight for each objective. Also, by accumulating the previous fairness gradient, CPT takes into account those subgroups that might be missing in the current mini-batch, which leads to a better fairness goal.

The moving average gradient of step $k$ is calculated with:

$$\bar{G}_j^k = \beta_j * \bar{G}_j^{k-1} + (1 - \beta_j) * \nabla_\theta \mathcal{L}_j(\theta) \tag{7}$$

where $\bar{G}_j$ and $\beta_j$ are the moving average gradient and the moving average weight for objective $j$.

### 3.4 Gradient Pruning in MGDA

In addition to refine MGDA with moving average gradient in section 3.3, we also intend to get a better common descent vector by denoising the gradient vector and lowering its dimension.

Since the parameters with higher values are more influential for the optimization process, we generate a mask based on parameters' magnitude and filter out the gradients of parameters with low magnitude. Unstructured magnitude pruning (Zhu & Gupta, 2017) is a commonly used technique for neural network pruning, which converts some of the parameters or weights with smaller magnitude into zeros. The prunning mask $\mathbf{m} \in \{0,1\}^d$ is generated with Alg. 1 and is updated every training iteration. The pruned gradient is calculated by:

---

**Algorithm 1:** Unstructured Magnitude Pruning

1 **Input** *Weight matrix $W$, Pruning ratio $p$*
2 Calculate the threshold $T = p * \max(|W|)$
3 **for** *each element $w_{i,j}$ in $W$* **do**
4    **if** $|w_{i,j}| \le T$ **then**
5       Set $m_{i,j}$ to 0
6    **else**
7       Set $m_{i,j}$ to 1
8 return Pruning Mask $\mathbf{m}$

---

$$\widetilde{G}_j = \mathbf{m} \odot \bar{G}_j \tag{8}$$

where $\bar{G}_j$ is the moving average gradient of objective $j$. With gradient pruning, we are able to accelerate the computation as well as get a better common descent vector.

### 3.5 Controllable Pareto Fairness-Accuracy Trade-off

In this section, we present our method CPT, whose detailed procedures are given in Alg. 2. First, CPT finds a starting point for multi-objective optimization that satisfies the constraint set by the reference vector $\boldsymbol{v}$. Then it jointly optimizes fairness $\mathcal{L}_{fair}$, accuracy $\mathcal{L}_{acc}$, and the constraint objective $\Psi(\boldsymbol{l}, \boldsymbol{v})$ to find the Pareto stationary solution in a certain region. Most of the time, the correction stage is applied in the beginning of optimization when $F(\theta)$ performs like a random classifier with high classification loss and low fairness loss. Also, when multi-objective optimization doesn't follow the reference vector, it helps the optimization to get back on track, which rarely happens (as shown in Fig. 4). The moving average gradients for accuracy $\bar{G}_{acc}$ and fairness $\bar{G}_{fair}$ are updated through the whole optimization process, while $\bar{G}_{kl}$ is updated only in the MOO stage where minimizing the KL divergence between reference vector and loss value vector becomes the third objective. In the MOO stage, CPT prunes the moving average gradient of each objective with the mask $\mathbf{m}$ and uses an existing multi-objective optimization algorithm to find a common descent vector. Following (Sener & Koltun, 2018), CPT also adopts the Frank-Wolf solver for solving the multi-optimization problem in Eq.( 6).

## 4 Experiments

In this section, we evaluate our proposed CPT from the following aspects: (1) Can CPT control the fairness-accuracy trade-off by precise reference vector following? (2) Can CPT generate more

---

**Algorithm 2:** Controllable Pareto Fairness-Accuracy Trade-off (CPT)

---

1  **Input** *dataset D, reference vector* $\boldsymbol{v} = (v_{fair}, v_{acc})$, *threshold* $\psi$, gradient moving average weight $W$

2  **Initialize** *model $F(\theta)$, FrankWolfSolver, optimizer $\rho$*

3  **for** *k=0,...,K* **do**

4     Update gradient mask $\mathbf{m}_k$ with Alg. 1

5     $\hat{y} = F(\theta, x_k, y_k, a_k)$

6     Compute $l_k^{acc}$ and $l_k^{fair}$ with Eq.( 2) and Eq.( 3)

7     Update $\bar{G}_{fair}$ and $\bar{G}_{acc}$ with Eq.( 7)

8     **if** $\Psi(\boldsymbol{l}, \boldsymbol{v}) > \psi$ **then**

        /* Correction Stage

9         **if** $l_{fair}/l_{acc} > v_{fair}/v_{acc}$ **then**

10            $G = \bar{G}_{fair}$

11         **else**

12            $G = \bar{G}_{acc}$

13     **else**

        /* MOO Stage

14         Update $\bar{G}_{kl}$ with Eq.( 7)

15         Get pruned gradient $\widetilde{G}_{fair}, \widetilde{G}_{acc}, \widetilde{G}_{kl}$ with Eq.( 8)

16         $\alpha = FrankWolfSolver(\widetilde{G}_{fair}, \widetilde{G}_{acc}, \widetilde{G}_{kl})$

17         Get common descent vector $G$ with Eq.( 9)

18     Update parameters: $\theta_{t+1} = \rho(\eta, G)$

---

diverse trade-off solutions between the two objectives? (3) Can the trade-off solution obtained by CPT generalize to unseen data? Specifically, Sec. 4.1 describes the experimental setting. Sec. 4.2 shows the superiority of CPT by comparing it with several state-of-the-art (SoTA) MOO methods. Sec. 4.3 presents a thorough ablation study to demonstrate the effectiveness of gradient moving average and gradient pruning.

## 4.1 EXPERIMENT SETTING

**Benchmarks.** We use Jigsaw dataset [1] to evaluate the performance of CPT on toxicity classification task and focus on race attribute as it has been proved to show the most significant bias over other attributes (Cheng et al., 2022). The statistic of different races is showed in Table 1. We utilize accuracy and equalized odd (EODD) (Hardt et al., 2016) as classification metric and fairness metric respectively to evaluate CPT. A classifier $F(\theta)$ satisfies EODD if the predicted outcome $Y$ is independent of the sensitive attribute $A$ conditioned on the label $Y$: $\mathbb{E}[F(X) \mid Y] = \mathbb{E}[F(X) \mid Y, A]$. A higher accuracy indicates better classification performance and a lower EODD value indicates there is less bias among different subgroups.

**Baselines.** We compare CPT with several baselines and SoTA MOO methods below:

(1) **Linearization** that directly optimizes a weighted sum of multiple objectives. i.e., $\mathcal{L} = \sum_{i=1}^{m} v_i \mathcal{L}_i$.

(2) **MGDA (Sener & Koltun, 2018) with diverse initialization**: We first provide MGDA with diverse initial solutions and then apply MGDA to solve bi-objective optimization with respect to each of them.

(3) **Pareto Multi-Task Learning (PMTL)** (Lin et al., 2019) generates solutions falling to different regions of the Pareto front by decomposing a multi-objective optimization problem into multiple sub-problems, each characterized by a distinct preference among those objectives.

| Subgroup | Label | |
|---|---|---|
| | Positive | Negative |
| White | 5636 | 5410 |
| Black | 3747 | 3050 |
| Latino | 313 | 497 |
| Asian | 183 | 224 |

Table 1: Statistics of training-set.

---

[1]https://www.kaggle.com/competitions/jigsaw-unintended-bias-in-toxicity-classification/data

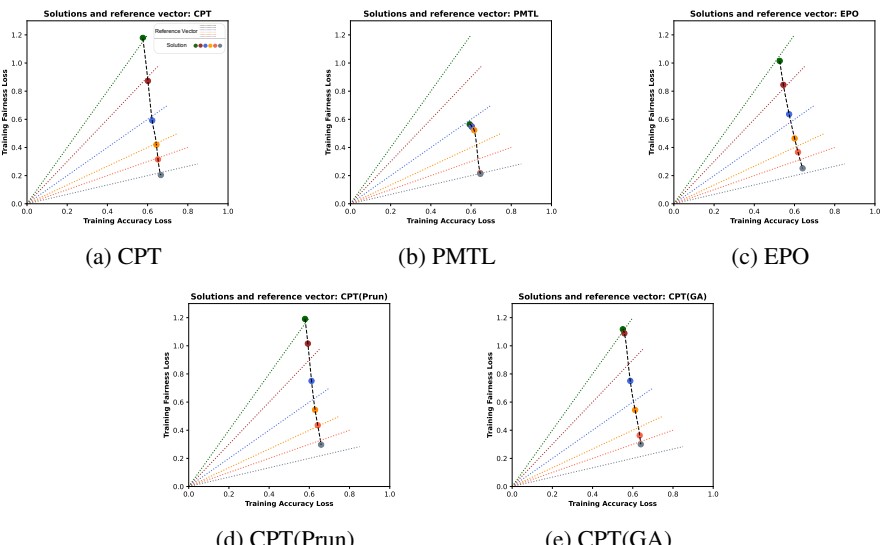

Figure 1: Fairness-accuracy trade-off solutions achieved by different methods using six reference vectors. **Among all methods, CPT is the best one whose solutions precisely follow the reference vectors**. Reference vectors from top to bottom are $(2, 1), (3, 2), (1, 1), (2, 3), (1, 2), (1, 3)$. The x-axis denotes the accuracy loss while the y-axis denotes the fairness loss on the training set.

(4) **Exact Pareto Optimization (EPO)** (Mahapatra & Rajan, 2020) combines multiple gradient descent with an elaborate projection operator to achieve convergence to the required Pareto solution.
(5) **CPT(GA)**: CPT with gradient moving average but without gradient pruning.
(6) **CPT(Prun)**: CPT with gradient pruning but without gradient moving average.

**Training details.** We apply sentence transformer (Reimers & Gurevych, 2019) as encoder and stack two fully connected layers as classification heads. We use an SGD optimizer with an initial learning rate of 0.01, which is decayed by a small constant factor of 0.8 until the number of epochs reaches a pre-defined value. All of the experiments are conducted on a 4090Ti GPU and run four random seeds for fair comparison. More details on the hyperparameters used in training can be found in the Appendix. A.3. In order to represent different trade-offs between fairness and accuracy, we set a diverse set of reference vectors: $\mathbb{V} = \{(2, 1), (3, 2), (1, 1), (2, 3), (1, 2), (1, 3)\}$, each of them is normalized by $\boldsymbol{v} = \frac{\boldsymbol{v}}{\|\boldsymbol{v}\|_1}$. By optimizing the loss function with the chosen reference vector (see Eq. 5), CPT can precisely control the trade-off between fairness and accuracy.

## 4.2 MAIN RESULTS

**Controllable Pareto trade-off by following reference vector.** In order to demonstrate the advantage of CPT, we compare it with two SoTA MOO methods: PMTL and EPO. As shown in Fig. 1, PMTL fails to generate diverse solutions with given reference vectors, and the solutions are mainly located in two regions. One possible explanation is that PMTL only uses reference vectors to determine initial solutions but lacks a principled method to follow them during the rest of the optimization process. While EPO achieves lower accuracy and fairness loss values than CPT for vectors with a preference for the accuracy objective (see $\boldsymbol{v} = (2, 1)$ and $\boldsymbol{v} = (3, 2)$), this advantage disappears on unseen data. The results in Fig. 2d indicate that EPO achieves worse fairness performance on the testing set for reference vectors $\boldsymbol{v} = (2, 1)$ and $\boldsymbol{v} = (3, 2)$, reflecting that EPO suffers from overfitting to training data. Furthermore, EPO fails to follow the reference vectors with higher fairness preference. This is because EPO uses a noisy stochastic gradient to determine the update direction for each step, which could be inaccurate as we discussed in Sec. 3.3, and thus the fairness performance is harmed. Fortunately, this challenge is successfully solved by CPT. Benefits from the pruning and moving average of gradients, CPT is able to precisely follow each reference vector.

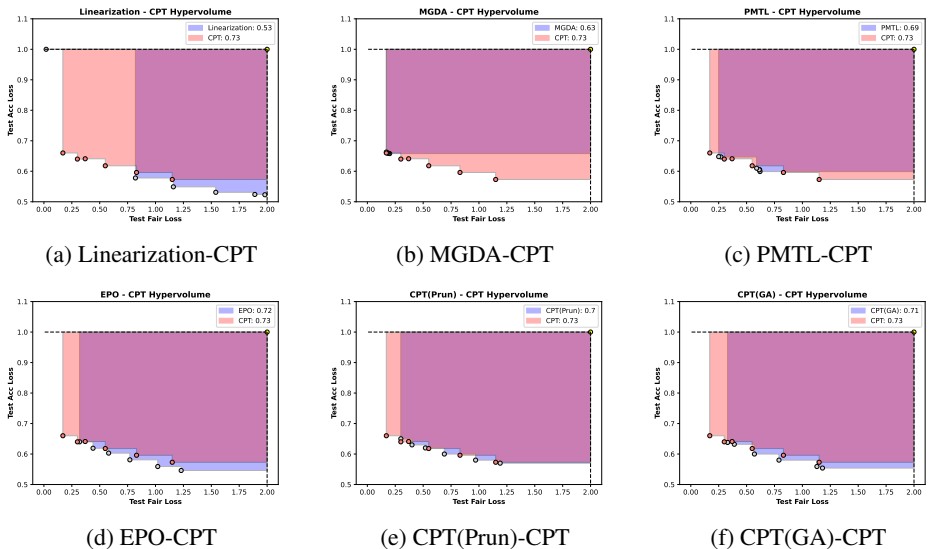

(a) Linearization-CPT      (b) MGDA-CPT      (c) PMTL-CPT

(d) EPO-CPT      (e) CPT(Prun)-CPT      (f) CPT(GA)-CPT

Figure 2: Hypervolume (test set) of the solutions achieved by different methods in the fairness-accuracy space. Numerical results are reported in Table 2. **CPT achieves the highest hypervolume, indicating the diversity of solutions** that provide different trade-offs.

**Evaluate diversity with fairness weighted hypervolume.** We evaluate CPT on the testing set and show the result in Table. 2 and Fig. 2. For a fair comparison, we apply the same reference point $(2, 1)$ for all methods. Hypervolume (Zitzler & Thiele, 1999) is a widely used metric in MOO. It calculates the area/volume of the resulting set of nondominated solutions with respect to a reference point to measure the diversity of these solutions (More details can be found in Appendix. A.2). In the experiment, the reference point is the worst-case result for each objective, i.e., the largest accuracy and fairness losses (see the yellow point on the right corner in Fig. 2). However, the original hypervolume metric neglects the difficulty of optimization for different objectives and treats them equally. For example, in our case, the fairness loss is harder to be optimized than the accuracy loss. In order to address this issue, we utilize a reference point that is more favorable to fairness. As shown in Table. 2, CPT achieves the best performance compared with other methods.

| **Method** | Linearization | MGDA | PMTL | EPO | CPT(GA) | CPT(Prun) | CPT |
|---|---|---|---|---|---|---|---|
| **Hypervolume** | 0.53 | 0.63 | 0.69 | 0.72 | 0.71 | 0.70 | **0.73** |

Table 2: Hypervolume (test set) of the solutions achieved by different methods in the fairness-accuracy space. CPT achieves the best hypervolume among all methods on the test set.

**Generalizable Pareto trade-off to unseen data.** When addressing the fairness-accuracy trade-off in real-world prediction problems, the resulting models are expected to work on training data meanwhile generalizing to unseen data. Hence, a reliable method should achieve a consistent fairness-accuracy trade-off on training and testing sets under the same reference vector. As shown in Table 3, only linearization and CPT exhibit this characteristic.

### 4.3 ABLATION STUDY

Here we study how the moving average and pruning of the objectives' gradients affect the performance. Comparing CPT(Prun) with CPT in Fig. 1, we find that there is a consistent increase of fairness loss for nearly all solutions, demonstrating that the gradient moving average technique can lead to a better fairness performance. On the other hand, when CPT(GA) removes the gradient pruning, the optimization process becomes more unstable, highlighting the importance of gradient pruning in stabilizing the optimization and determining a more accurate descent direction.

| Method | Accuracy↑ | | | | | | EODD↓ | | | | | |
|---|---|---|---|---|---|---|---|---|---|---|---|---|
| | (2,1) | (3,2) | (1,1) | (2,3) | (1,2) | (1,3) | (2,1) | (3,2) | (1,1) | (2,3) | (1,2) | (1,3) |
| Linearization | **+0.48** | +0.07 | 74.35 | -0.15 | -0.30 | -23.55 | +1.54 | +0.84 | 5.86 | -0.73 | -0.72 | **-5.75** |
| MGDA | -1.13 | -1.13 | **71.96** | -0.26 | -0.33 | -0.51 | +1.28 | +1.28 | **8.20** | +2.68 | +4.30 | +1.78 |
| PMTL | **+0.67** | -4.09 | 72.71 | -0.93 | -0.54 | -0.21 | +0.37 | -0.34 | 4.90 | +4.37 | **-1.02** | -0.32 |
| EPO | **+0.31** | +0.80 | 73.63 | +0.03 | -0.25 | -0.54 | -1.77 | -1.39 | 6.69 | -1.61 | **-1.85** | -1.15 |
| CPT(GA) | +0.33 | **+0.46** | 73.48 | -0.52 | -1.52 | -1.26 | **-1.44** | +0.16 | 5.64 | +0.12 | +1.81 | +1.24 |
| CPT(Prun) | **+1.41** | +1.26 | 71.55 | -0.80 | -1.11 | -2.01 | +6.87 | +5.26 | **1.74** | +0.38 | +2.65 | +1.83 |
| CPT(Ours) | **+1.11** | +0.44 | 72.09 | -0.70 | -1.31 | -2.29 | +4.39 | +1.81 | 3.47 | -0.66 | -0.89 | **-0.92** |

Table 3: Accuracy and EODD (fairness) on the test set. The results for reference vector $v = (1,1)$ are reported in their original values, while the results for the other five reference vectors are differences from metrics achieved at $v = (1,1)$. For each method, the best accuracy and fairness among the six reference vectors are highlighted by **bold**. **CPT's fairness and accuracy on the test set successfully match the reference vectors** while other methods failed.

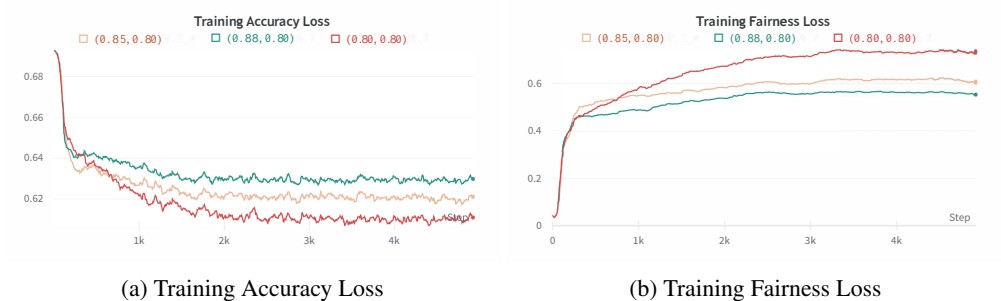

(a) Training Accuracy Loss        (b) Training Fairness Loss

Figure 3: Moving average weights $\beta_{fair} \in \{0.80, 0.85, 0.88\}$ applied to the fairness gradients when using reference vector $v = (1,1)$. While the solution associated with $\beta_{fair} = 0.85$ is the closest to $v$, increasing (decreasing) $\beta_{fair}$ introduces a bias further minimizing the fairness (accuracy) loss.

Then we explore how different moving average weights affect the optimization. We set reference vector to $v = (1,1)$, fix the weight for accuracy ($\beta_{acc} = 0.80$), and apply different weights ($\beta_{fair} = \{0.88, 0.85, 0.80\}$) for fairness. The results in Fig. 3 indicate that increasing the moving average weight could provide us with solutions close to reference vectors. For example, when $\beta_{acc} = 0.80, \beta_{fair} = 0.85$, the resulting solution better follows the reference vector. On the contrary, if $\beta_{fair}$ decreases, CPT will generate solutions with more preference for accuracy.

## 5 CONCLUSIONS

In this paper, we present CPT, a method for controllable pareto fairness-accuracy trade-off. CPT provides two techniques to refine the application of gradient-based multi-objective optimization method in fairness-accuracy trade-off. First, CPT applies moving average gradient instead of stochastic gradient for each objective, which stabilizes the training process and results in better fairness performance. Second, CPT generates a mask based on parameter magnitude to prune the gradient, the denoised low dimensional gradient benefits MOO by providing a more precise common descent vector. We evaluate CPT on real-world dataset and show its advantage in both optimization process and test results. In feature work, we would like to explore how to get a set of Pareto stationary solutions near the reference vector instead of a single solution for each vector.

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

# A APPENDIX

## A.1 COMMON DESCENT VECTOR

The common descent vector in multi-objective optimization can be defined as:

$$\nabla_\theta \mathcal{L}(\theta) \triangleq \sum_{i=1}^{m} \alpha_i \nabla_\theta \mathcal{L}_i(\theta) \tag{9}$$

where $\nabla_\theta \mathcal{L}_i(\theta)$ indicates the gradient of the $i$-th objective and $\alpha_i$ is the weight for the $i$-th objective. Considering the case of two objectives, the optimization problem could be defined as $\min_{\alpha \in [0,1]} \left\| \alpha \nabla_\theta \mathcal{L}_1(\theta) + (1-\alpha)\nabla_\theta \hat{\mathcal{L}}_2(\theta) \right\|_2^2$. Then, the analytical solution for $\alpha$ is:

$$\alpha = \frac{(\nabla_\theta \mathcal{L}_2(\theta) - \nabla_\theta \mathcal{L}_1(\theta))^T * \nabla_\theta \mathcal{L}_2(\theta)}{\left\| \nabla_\theta \mathcal{L}_1(\theta) - \nabla_\theta \mathcal{L}_2(\theta) \right\|^2} \tag{10}$$

When it comes to multiple objectives, the calculation of common descent vector still relies on the inner product.

## A.2 HYPERVOLUME

Hypervolume is a valuable metric in multi-objective optimization that measures the quality of a set of solutions by quantifying the objective space they cover. The hypervolume metric can be defined as follows: Given a set of points $P \subset \mathbb{R}^n$ and a reference point $r \in \mathbb{R}_+^n$, the hypervolume of $\mathbb{R}$ is measured by the region of non-dominated points bounded above by $r$:

$$HV(P) = \text{VOL}\left( \left\{ s \in \mathbb{R}_+^n \mid \exists p \in P : (p \preceq s) \wedge (s \preceq r) \right\} \right) \tag{11}$$

In the bi-optimization problem, it can be represented by the area of the polygon bounded by the solution set and reference point.

## A.3 IMPLEMENTATION DETAILS

The version of Sentence Transformer we use is paraphrase-MiniLM-L3-v2 [2]. And the classifier is consist of two fully connection layers with size (384, 384) and (384,1). The output of the final layer is the probability of being toxic. We utilize SGD with 0.9 momentum. The learning rate is set to 0.01 initially and decreases every epoch with 0.8 decay rate. The number of epoch is 40 and the batch size is set to 128. As for hyperparameters related to our method, we set the threshold $\psi$ to be 0.002. The moving average weights are provided in Table 4.

Table 4: Moving average weights for reference vector.

| Reference Vector | Fairness Weight | Accuracy Weight |
|:---:|:---:|:---:|
| (2,1) | 0.50 | 0.50 |
| (3,2) | 0.20 | 0.16 |
| (1,1) | 0.15 | 0.20 |
| (2,3) | 0.12 | 0.15 |
| (1,2) | 0.10 | 0.25 |
| (1,3) | 0.10 | 0.35 |

---

[2]https://huggingface.co/sentence-transformers/paraphrase-MiniLM-L3-v2

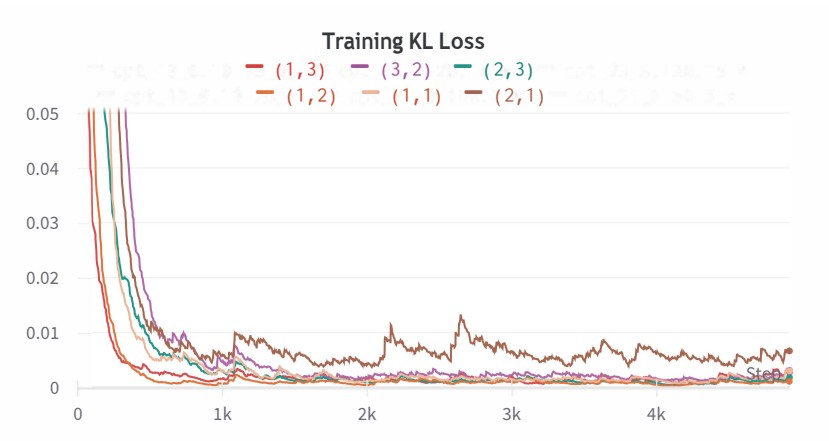

Figure 4: Training KL divergence loss: The KL loss decreases from correction stage to MOO stage and converges at the end of training, which indicates the optimization process follows the reference vector very well.

