# OpenReview forum: "Controllable Pareto Trade-off between Fairness and Accuracy"
_ICLR.cc/2024/Conference — ICLR 2024 Conference Withdrawn Submission_

### Official Review · Reviewer_MTak · 2023-10-31

**Soundness:** 3 good
**Presentation:** 3 good
**Contribution:** 2 fair
**Rating:** 3
**Confidence:** 3

**Summary:**

In this paper, the authors focus on the problem of controlling the trade-off between accuracy and fairness that one encounter in modern machine learning problems. To this end, authors introduce an algorithm that (1) optimize fairness and accuracy objectives simultaneously (2) maintain a user specified preference over fairness and accuracy objectives. To achieve this, the authors suggest using a moving average	of gradients and gradient magnitude pruning that result in a better estimate of a conflict avoiding direction, and using an additional objective that enforces the compliance of the learned model to a user specified preference. The authors provide empirical results that compares the proposed method with existing Pareto front learning baselines, and show that the proposed algorithm recover a Pareto front that have good properties like better spread and larger hyper-volume.

**Strengths:**

* The paper propose and algorithm that tries to balance between simultaneously optimizing two objectives, while maintaining a given preference, which is an interesting idea.

* The paper provide some empirical evidence for the efficacy of the proposed method over existing methods that are designed to recover the Pareto front with good properties like better spread and larger hyper-volume, while maintaining the required preference.

**Weaknesses:**

* While the authors list “utilize the moving average of stochastic gradients to approximate the full gradients” as a contribution of the paper, this idea have been already proposed in prior work like [1].

* Some work like [2] which also try to have a balance between simultaneously optimizing multiple objectives while maintaining a given preference is not compared and contrasted in the paper.

* The definition of Pareto stationary used in this paper seems to be different from the usual definition used in the literature [1, 2, 3].

* Given that the paper provide only empirical results to validate the proposed method, considering only one benchmark to compare different methods seems limited evaluation of the proposed method.

Minor comments:

* “Pareto frontier”, TPR, FPR are used in paper without defining the terms.

* Regarding Figure 1, it's better to have all Pareto fronts for each method in the same figure for better comparison.

[1] Fernando, H.D., Shen, H., Liu, M., Chaudhury, S., Murugesan, K. and Chen, T., 2022, September. Mitigating gradient bias in multi-objective learning: A provably convergent approach. In The Eleventh International Conference on Learning Representations.

[2] Momma, M., Dong, C. and Liu, J., 2022, June. A multi-objective/multi-task learning framework induced by pareto stationarity. In International Conference on Machine Learning (pp. 15895-15907). PMLR.

[3] Liu, S. and Vicente, L.N., 2021. The stochastic multi-gradient algorithm for multi-objective optimization and its application to supervised machine learning. Annals of Operations Research, pp.1-30.

**Questions:**

* How does the momentum technique introduced here defers from [1], which also use a momentum based technique to ensure reduced bias for the stochastic estimate of MGDA direction?

* What are the benefits of the proposed reference following method over the method provided in Yang et al. 2021 (reference as appearing in the paper) ?

* Can the authors provide more empirical evaluation of the method comparing other methods for other benchmarks?

---

### Official Review · Reviewer_8pma · 2023-11-01

**Soundness:** 1 poor
**Presentation:** 2 fair
**Contribution:** 2 fair
**Rating:** 3
**Confidence:** 3

**Summary:**

This paper proposes a method for multi-objective optimization (MOO), named Controllable Pareto Trade-off (CPT), which can trade off multiple objectives based on reference vectors.
Specifically, it proposes three major components in the method --

1. Use reference vectors to guide the optimization process. The KL-divergence with the reference objective serves as an additional objective along with the fairness and accuracy objectives.

2. Use the moving average of the stochastic gradient to stabilize the training process.

3. Use gradient pruning to accelerate computation.

The proposed method is applied to trade off fairness and accuracy for machine learning problems. Experiments are conducted on the Jigsaw dataset to show the effectiveness of the proposed method.

**Strengths:**

The method proposed by the paper is easy to follow. However, I believe some definitions and notations are inaccurate or incorrect. See weaknesses.

**Weaknesses:**

1. Section 3.1 Definitions are incorrect or unclear

a) Definition 1-2) is incorrect. This is the definition for Pareto optimal instead of Pareto stationary, these two concepts are not equivalent unless some assumptions are imposed.

b) Definition 3 is incorrect. In Appendix A.1, it is defined as the convex combination of gradients, which cannot guarantee update along the direction will lead to the descent of all objectives.

2. The motivation for using gradient pruning is unclear.
After pruning, how can your algorithm guarantee a common descent direction? Justification needs to be provided on this aspect, otherwise, most claims and motivations for this paper are not supported or contradictory.


3. Some references are missing.

a) Using the moving average of stochastic gradients for MOO is not new, see [3].

b) Some recently proposed MOO methods are not being discussed or compared. See the references below.

4. Experiments are insufficient.

In section 4.1, the authors claim to compare with SOTA MOO methods, however, the most recent method compared is in 2020. Below I listed a few methods that are more recently proposed.

[1] Liu, et al. "Conflict-Averse Gradient Descent for Multi-task Learning" NeurIPS, 2021

[2] Zhou, et al. "On the convergence of stochastic multi-objective gradient manipulation and beyond" NeurIPS, 2022

[3] Fernando, et al. "Mitigating gradient bias in multi-objective learning: A provably convergent stochastic approach" ICLR, 2023

[4] Chen et al. "Three-way trade-off in multi-objective learning: Optimization, generalization and conflict-avoidance" NeurIPS, 2023

[5] Xiao et al. "Direction-oriented Multi-objective Learning: Simple and Provable Stochastic Algorithms" NeurIPS, 2023

**Questions:**

### Major

1. In the abstract, why the "MOO methods usually lack precise control of the trade-offs. They rely on the full gradient per objective"?

2. Some important notations are not defined clearly.

a) In Section 3.4, what is $d$?
It seems to be the dimension of the model parameters or gradients.

b) In Algorithm 1, what is $\max (|W|)$?
From the algorithm, it seems this is the maximum of the absolute value of the elements in $W$. This is unclear as sometimes $|\cdot|$ can also be used as cardinality.



### Minor
1. Section 3.2 before Eq.(5),

"between een" -> "between"

remove "\in"

2. Section 5
"In feature work" -> "In future work"

---

### Official Review · Reviewer_whui · 2023-11-06

**Soundness:** 2 fair
**Presentation:** 1 poor
**Contribution:** 2 fair
**Rating:** 3
**Confidence:** 4

**Summary:**

This paper focuses on optimizing the fairness accuracy trade-offs and aiming to control the trade-offs manually. Thus this paper proposes a method called Controllable Pareto Trade-off (CPT) for achieving diverse and controllable trade-offs between fairness and accuracy for machine learning models. CPT allows precisely controlling the trade-off by following pre-defined reference vectors in the objective space. The claimed contributions of this paper are:
1) using moving averages of stochastic gradients to approximate full gradients, reducing noise and missing subgroups, and finding the common descent direction without missing subgroups
2) pruning gradients to reduce dimensionality and enable a more accurate estimation of the common descent direction.
3. Experiments on toxicity classification show CPT can achieve more diverse Pareto solutions compared to prior methods.

**Strengths:**

1. The multi-objective optimization typically results in an uncontrollable and unbalanced fairness-accuracy trade-off, which is a real problem in practice.
2. The paper is easy to follow. The problem definition and method description are detailed.

**Weaknesses:**

1. The motivation of this paper is not clear, and the proposed method is not well-motivated.
- 1.1 What are the real-world scenarios in which controllable fairness-accuracy trade-offs are needed?
- 1.2 The controllable fairness-accuracy trade-offs may incur ethical issues, such as generating biased outcomes for certain groups of people. Based on this, I think this paper needs further ethical review.
- 1.3 To achieve fairness in the downstream task, we only need a final fair model. How is the proposed model applied to real tasks?
2. The technical contribution is limited and not sound to me. The adopted techniques are all from previous work, such as MGDA and gradient pruning. I think directly adopting an existing technique to a new scenario is acceptable, but combined with Weakness 1, this is questionable.
3. The presentation of this paper is poor.
- 3.1 The format of the definition is strange and not formal for a research paper.
- 3.2 Figure 1 is hard to read since the title and axes are too small. Figure 2 has the same issue.
4. The experiment is not convincing at all.
- 4.1 Only one dataset is used. This is not enough to evaluate the effectiveness of the proposed method. For the fairness domain, the more commonly used datasets such as tabular data (folktable, German Credit, COMPAS), and image data (CelebA) are not discussed and experimented on.
- 4.2 Even for Jigsaw, the backbone is also limited. Using a BERT as an encoder and two-layer MLP for classification does not investigate the proposed method well. More backbones such as end-to-end language models should be considered.
- 4.3 Why only report the loss in Figures 1 and 2? The accuracy metric of the accuracy-fairness trade-off curve should also be presented and investigated.
5. Minor Question:
- 5.1 Please show more detail of Equation (3). Since this loss function seems non-differentiable, the paper should present how this loss integrates into the overall loss or is approximately integrated into it.
- 5.2 In the abstract, this paper said “combining the two objectives can result in mediocre or extreme solutions” but introduces a third objective in the loss function “the reference vector and then includes the discrepancy between the reference and the two objectives as the third objective in the rest training”. Is this contradictory?


This paper does not meet the standards for acceptance to ICLR in its current form. For now, I would recommend rejection.

**Questions:**

Please address my concerns in the Weakness.

**Details Of Ethics Concerns:**

The controllable fairness-accuracy trade-offs may incur ethical issues, such as generating biased outcomes for certain groups of people. Based on this, I think this paper needs further ethical review.